# The Cellular Microbiome of Visceral Organs: An Inherent Inhabitant of Parenchymal Cells

**DOI:** 10.3390/microorganisms12071333

**Published:** 2024-06-29

**Authors:** Xiaowei Sun, Hua Zhang, Xiao Zhang, Wenmin Gao, Caiyun Zhou, Xuanxuan Kou, Jingxin Deng, Jiangang Zhang

**Affiliations:** Pathology Institute, School of Basic Medical Sciences, Lanzhou University, Lanzhou 730000, China; hzhang2020@lzu.edu.cn (H.Z.); zhangxiaosj7@gmail.com (X.Z.); gaowm19@lzu.edu.cn (W.G.); 18919802202@163.com (C.Z.); kouxx20@lzu.edu.cn (X.K.); dengjx21@lzu.edu.cn (J.D.)

**Keywords:** cellular microbiome, 16S rRNA gene, LPS, LTA, inherent inhabitants

## Abstract

The cell is the basic unit of life. It is composed of organelles and various organic and inorganic biomolecules. Recent 16S ribosomal ribonucleic acid (16S rRNA) gene sequencing studies have revealed the presence of tissue bacteria in both tumor and normal tissues. Recently, we found that the liver microbiome resided in hepatocytes. Here, we further report on the cellular microbiome in the parenchymal cells of visceral organs as inherent inhabitants. We performed 16S rRNA gene sequencing on visceral organs of male adult Sprague Dawley (SD) rats, pregnant rats, newborn rats, and fetuses and placentas; then, we performed fluorescence in situ hybridization and immunofluorescence in visceral organs. Furthermore, we performed Western blotting on nuclear and cytoplasmic extractions of visceral organs of SD rats and cell lines HepG2, Huh-7, Hepa1-6, and HSC-T6. A high abundance of 16S rRNA gene was detected in the visceral organs of male adult, pregnant, newborn, and fetal rats as well as their placentas. The number of operational taxonomic units (OTUs) of visceral bacteria was higher than that of the feces and ileum bacteria. Bacterial 16S rRNA, lipopolysaccharide (LPS), and lipoteichoic acid (LTA) were found in the parenchymal cells of visceral organs, as well as in HepG2, Huh-7, HSC-T6, and Hepa1-6 cells. LPS consistently appeared in the nucleus of cells, while LTA was mainly found in the cytoplasm. In conclusion, the cellular microbiome is an intrinsic component of cells. Gram-negative bacteria are located in the nucleus, and Gram-positive bacteria are located in the cytoplasm. This differs from the gut microbiome and may be inherited.

## 1. Introduction

Microorganisms are a natural part of the systems that maintain stability and quality of life for humans and animals [1]. However, microorganisms also cause many diseases [2,3,4,5]. As a major threat to human health, microorganisms can lead to infection and cause infectious diseases [6]. The gut microbiome has recently become a focal point for biological and medical researchers [7]. This interest is largely due to the revolutionary development of high-throughput 16S ribosomal ribonucleic acid (16S rRNA) gene sequencing. With next-generation sequencing, trillions of bacteria can be detected as operational taxonomic units (OTUs) and annotated. The most valuable use of gut microbiome identification is to determine its relationship with various intestinal diseases [8,9], as well as with various non-infectious diseases of visceral organs [7,8,10,11,12,13,14,15,16,17,18,19,20,21,22,23]. According to the research in this field, the gut microbiome has inevitably become the leading cause of many diseases. However, it is still unknown whether changes in intestinal bacteria are the reason or result of these chronic diseases. The most popular opinion tends to favor the former. Moreover, the mechanisms by which the gut microbiome influences the inner environment and causes diseases are still a mystery. The current thought is that they are caused by gut leakage [24,25,26,27,28,29,30].

The cell is considered the primary functional unit of life [31]. In the cellular and molecular biology systems of the cell, because there is no electron microscopic evidence of bacteria, exogenous bacteria are excluded, although oncogenes, mitochondrial and chloroplast DNA, and viral particles are present. Intracellular bacteria are always found in phagocytes. In fact, there are still a large number of particles in cells that are not well recognized. Recently, two leading studies further reported bacterial 16S rRNA genes in tumor tissues. Furthermore, they also confirmed the intracellular existence of bacteria by in situ detection using immunohistochemistry or electronic microscopy. Nejman et al. collected 1526 tumors and adjacent normal tissue of seven cancer types and showed the existence of a tumor-specific intracellular microbiome [32]. Fu et al. showed that tumor-resident microbiota played an important role in promoting cancer metastasis [33]. These results further confirmed the important role that the gut microbiome may have in the development of tumors [34,35]. This also raises important concerns. Where are tumor intracellular bacteria from? In addition, if they are derived from gut microbiota, how and when did they enter the cells?

The finding of tumor intracellular bacteria changed the traditional thinking on the cell structure. Based on the discovery of bacterial DNA in the liver, adipose tissue, heart, brain, muscle [36], and leukocyte and platelet fractions from the whole blood of healthy humans (but not the plasma) [37,38], the concept that gut microbiota reach other organs from the circulation after gut injury has been challenged. Recently, through the detection of 16S rRNA genes, lipopolysaccharide (LPS), and lipoteichoic acid (LTA) by means of next-generation sequencing, we found that liver microbiomes are hidden inhabitants of hepatocytes [39]. We thus hypothesized that the cellular microbiome should also be an inhabitant of normal parenchymal cells. In this article, by investigating visceral organs of Sprague Dawley (SD) rats, newborn rats, placenta, and cultured cell lines, we suggest that bacteria are inherent inhabitants of normal parenchymal cells.

## 2. Materials and Methods

### 2.1. Animals

Male adult (*n* = 6, body weight = 260 to 360 g), pregnant (*n* = 5, body weight = 300 to 450 g), newborn (*n* = 6), and fetal (*n* = 5) SD rats were screened in this study. The study protocols complied with the laboratory animal—guideline for ethical review of animal welfare (GB/T 35892—2018) and were approved by the Medical Ethics Committee of Lanzhou University (jcyxy20190302).

### 2.2. Cells and Culturing

Human hepatoma HepG2 (#CL-0103), Huh-7 (#CL-0120), and mouse hepatoma Hepa1-6 (#CL-0105) cells (Procell Life Science & Technology Co., Ltd. Wuhan, China) and rat hepatic stellate HSC-T6 cells (#KCB200703YJ) (Kunming Cell Bank (Chinese Academy of Sciences, China)) were cultured with Dulbecco’s modified Eagle’s medium (DMEM) supplemented with 10% fetal bovine serum (FBS, Sigma-Aldrich, St. Louis, MO, USA) and 100 U/mL penicillin/streptomycin at 37 °C in a 5% CO_2_ incubator.

### 2.3. Sample Collection and Contamination Avoidance

Amicrobic tissues were sampled in our laboratory as previously reported [39]. Adult animals were anesthetized with amicrobic pentobarbital sodium (5 mg/100 g body mass) and sacrificed through portal vein blood drainage. The liver, spleen, kidney, pancreas, heart, lung, skeletal muscle, and brain, followed by the jejunum (20 cm distal to the duodenum), ileum (20 cm proximal to the ileocecum), and colon, were sampled successively. Briefly, after sacrifice through portal vein blood drainage, the fetus and placenta were collected. The spleen, kidney, and pancreas were then separated after clamping the splenic and renal arteries and veins. The heart and lung lobes were separated following the opening of the thoracic cavity and closure of the superior and inferior vena cava. The adductor major muscle was subsequently separated, after which the skull was quickly opened, and the brain tissue was completely separated from it. Finally, stomach and intestinal tissues were sampled. Newborn rats were also subjected to skin sterilization and contamination avoidance procedures before being sacrificed through decapitation. The entire intestinal tissues of newborn rats were collected together with their contents and were not separated further due to their small size. Milk clots were pushed out from the stomach followed afterwards by skin tissue sampling. All samples were stored at −80 °C after being rapidly frozen in liquid nitrogen or fixed with 4% paraformaldehyde.

### 2.4. Bacterial 16S rRNA Gene Sequencing

Total DNA was extracted using a Magnetic Soil and Stool DNA Kit (#DP812; TianGen Corp., Beijing, China, https://www.tiangen.com/, accessed on 25 November 2020) according to the kit protocol. We used two rounds of tailed PCR for 16S amplification and sequencing [39]. The bacterial primers were as follows: 338F: 5′-ACTCCTACGGGAGGCAGCA-3′; 806R: 5′-GGACTACHVGGGTWTCTAAT-3′ [40].

Sequencing was performed on an Illumina HiSeq 2500 (male adult SD rats, data available in PRJNA831335) or an Illumina NovaSeq 6000 (pregnant, newborn, or fetal rats, PRJNA857281 and PRJNA857328) at Biomarker Technologies Co, Ltd. (Beijing, China) (www.biomarker.com.cn, accessed on 25 November 2020 and 14 July 2021, BMK200916–AC763–0101 and BMK210512–AJ541–ZX01-0101). The sequencing length was 350–450 bp.

### 2.5. Immunofluorescence Assays

Frozen tissue sections were treated with 4% paraformaldehyde for 50 min at room temperature (RT) and with 0.3% Triton X-100 (#T8200; Solarbio, Beijing, China) and 3% bovine serum albumin (BSA; B2064-50G; Sigma Germany, Munich, Germany) for 50 min at 37 °C. After incubation with primary antibodies overnight at 4 °C, sections were treated with a secondary antibody for 1 h at 37 °C and counterstained with DAPI (0.5 μg/mL; Ex/Em = 364/454, #BL105A; BioSharp, Hefei, China) at room temperature for 7 min. We observed and recorded images using a Nikon-ECLIPSE 80i/DS-Ri2/NIS-Elements D microscope (Nikon, Tokyo, Japan). The primary antibodies used were lipopolysaccharide (LPS) core, monoclonal antibody [mAb] WN1 222-5, #HM6011, 1:800 dilution and lipoteichoic acid (LTA), mAb 55, #HM2048, 1:800 dilution; both antibodies were from Hycult Biotech, Uden, The Netherlands. The secondary antibody used was DyLight 488 goat anti-mouse immunoglobulin G [IgG], Ex/Em = 493/518 nm, #AMJ-AB2004, 1:800 dilution, AmyJet Scientific Inc., Wuhan, China. Paraformaldehyde-fixed *Staphylococcus* and *Escherichia coli* were used as Gram-positive and Gram-negative controls, respectively.

### 2.6. 16S rRNA Fluorescent In Situ Hybridization

Frozen tissue slides were pretreated with 4% paraformaldehyde, 0.3% Triton X-100, 1% lysozyme, and DEPC water according to the instructions of the EUB338 FISH Probe Kit (#20 μM; FBPC-10; Creative Bioarray, Shirley, NY, USA). Bacteria in logarithmic growth were used as a positive control and pretreated with 4% paraformaldehyde, 0.01 M HCl, and lysozyme. During hybridization, slides were fixed with 4% paraformaldehyde for another 15 min at RT, after which we treated them with diethyl pyrocarbonate (DEPC) for 10 min and incubated them with BSA (3%) at 37 °C for 2 h. BSA was then discarded, and fluorescein isothiocyanate (FITC)-labeled probes (EUB338 GCTGCCTCCCGTAGGAGT) and a non-specific complement probe (nEUB338 CGACGGAGGG CATCCTCA; #FBPC-13; Creative Bioarray) were hybridized with the bacteria overnight at 42 °C. We counterstained slides with DAPI antifade solution (using EUB338 FISH Probe Kit) for 10 min and examined them under a Nikon fluorescence microscope.

### 2.7. LPS and LTA Western Blotting

We separated cellular cytoplasmic and nuclear compartments using a Minute™ Cytosolic and Nuclear Extraction Kit for Frozen/Fresh Tissues (#NT-032; Invent, Beijing, China) and Minute™ Cytoplasmic and Nuclear Fractionation Kit for Cells (#SC-003; Invent, Beijing, China) according to the protocol of the kits. The bicinchoninic acid (BCA) method was used to measure protein concentrations. Protein samples were separated by sodium dodecyl sulfate polyacrylamide gel electrophoresis (SDS-PAGE). The primary antibodies used were LPS (1:800; HM6011; Hycult), LTA (1:800; #HM2048; Hycult), Lamin-B1 (1:2000, #ab16048; Abcam, Cambridge, UK), and GAPDH (1:5000; #YM3215; Proteintech, Chicago, IL, USA). The secondary antibodies were horseradish peroxidase (HRP)-labeled goat anti-rabbit (1:5000, #RS0002; ImmunoWay Biotechnology Co., Plano, TX, USA) and HRP-labeled goat anti-mouse (1:5000; #RS0001; ImmunoWay). Protein bands were visualized using an electrochemiluminescence (ECL) kit (Super ECL Detection Reagent, Yeasen Biotechnology Co., Ltd., Shanghai, China) and ImageJ software v6.0. The gel and blot images were captured using a Chemiluminescence imaging system (FUSION SOLO6S EDGE, VILBER, FRANCE).

### 2.8. Statistical Analysis

All data were expressed as mean ± standard deviation. We conducted all statistical analyses using SPSS software v19.0 (IBM Corp., Armonk, NY, USA). An independent sample *t* test and one-way analysis of variance (ANOVA) were used for mean comparisons. Differences were considered statistically significant at *p* < 0.05.

## 3. Results

### 3.1. Visceral Bacteria in Male Adult SD Rats

A large number of 16S rRNA genes were detected in the visceral organs. The number of OTUs (1432.00 ± 39.87) was higher than that of the feces and ileum bacteria (1009.00 ± 5.66), *p* < 0.05 (independent *t* test) (Figure 1A). Nearly all the OTUs were shared across organs and intestinal contents (Figure 1B). However, the dominant bacteria varied. The phyla Firmicutes and Bacteroidetes and genera *Lactobacillus* were dominant in intestinal contents, while the phyla Proteobacteria and genera *Cupriavidus* were dominant in visceral organs (Figure 1C,D). *Halomonas*, *Ralstonia*, *Lactobacillus*, *Dietzia*, *Sphingomonas*, uncultured bacterium *Enterobacteriaceae*, uncultured bacterium subgroup 6, *Ochrobactrum*, and *Brevibacterium* were the other top 10 bacteria in visceral organs, and *Romboutsia*, uncultured bacterium f *Muribaculaceae*, uncultured bacterium f *Lachnospiraceae*, *Candidatus Arthromitus*, uncultured bacterium f *Prevotellaceae*, *Rodentibacter*, uncultured bacterium f *Ruminococcaceae*, *Gram-negative bacterium cTPY-13*, and *Blautia* were the other top 10 dominant intestinal bacteria. Except for the brain exclusive genera *Spirosoma*, and the kidney exclusive genera uncultured bacterium p *Patescibacteria*, there were no exclusive bacteria in other organs or intestinal contents. Among the top 10 dominant bacterial phyla in visceral organs, the abundance of Gram-negative bacterial phyla Proteobacteria, Bacteroidetes, Acidobacteria, Cyanobacteria, Gemmatimonadetes, Epsilonbacteraeota, Chloroflexi, and Verrucomicrobia was significantly higher than that of Gram-positive bacterial phyla Firmicutes and Actinobacteria (64.24 ± 6.65% vs. 34.17 ± 6.87%, *p* < 0.01). Furthermore, Gram-positive bacteria were significantly higher than Gram-negative bacteria in intestinal contents (77.74 ± 19.09% vs. 21.70 ± 18.86%, *p* < 0.01). Additionally, Gram-positive bacteria were significantly higher in intestinal contents than in visceral organs (*p* < 0.001), and Gram-negative bacteria were significantly higher in visceral organs than in intestinal contents (*p* < 0.001).

The alpha diversity (species richness and evenness) of the visceral microbiome was significantly higher than that of the intestinal microbiome (*p* < 0.05), and there was no difference between visceral organs (*p* > 0.05), with the diversity of the liver microbiome exceptionally lower than that of the heart, lung, and pancreas (Figure 2A). Based on PCA, the beta diversity of the visceral microbiome was significantly lower than that of the intestinal microbiome, *p* < 0.05 (Figure 2B). The visceral bacteria showed higher similarity and consistent abundance.

Among 613 annotated genera visceral bacteria, 404 types of bacteria were shared across organs, including the top 10 bacteria (Appendix A). Other bacteria were shared among the different organs, except for very few that were exclusive to the brain, heart, lung, kidney, and spleen (Appendix A). Within an individual, approximately 15–40% (23.23% ± 7.91%) of the visceral bacteria were shared with other organs and 8–11% (9.56% ± 1.37%) were exclusive to specific organs. This result suggested that visceral bacteria were structurally different from intestinal bacteria, although the bacteria were shared.

In each type of organ, the abundance of dominant bacteria varied across individuals (Appendix A). For each individual, the abundance of dominant bacteria varied across organs and was usually different from the bowel microbiome (Appendix A). Although a large number of species were shared, each organ had exclusive bacteria with a low abundance (from 1 to 672) (Appendix A).

According to a Kyoto Encyclopedia of Genes and Genomes (KEGG) analysis, the functional composition of the visceral bacteria genomes across organs was the same. These genomes were in lower number in “genetic information processing” and “environmental information processing” and in higher number in “metabolism and cellular processes” than those of the intestinal bacteria (Appendix A). “Fermentation and chemoheterotrophy” was lower than in intestinal bacteria, and “aerobic chemoheterotrophy”, “nitrate reduction”, “hydrocarbon degradation”, “predatory or exoparasitic”, “aromatic compound degradation”, “aromatic hydrocarbon degradation”, and “ureolysis” were higher than in intestinal bacteria (Appendix A). The intestinal bacteria showed a high abundance of “anaerobic”, “Gram-positive”, “facultatively anaerobic” (jejunum and ileum), and “mobile elements” (jejunum and ileum), while the visceral bacteria were “potentially pathogenic”, “form biofilms”, “Gram-negative”, “aerobic”, and “stress tolerant” (Appendix A). This result suggested that the visceral bacteria were different from intestinal bacteria in genome functional composition.

We further conducted immunofluorescence (IF) and Western blotting using antibodies against bacterial LPS and LTA to detect Gram-negative and Gram-positive bacteria, respectively. Both LPS and LTA were found in the parenchymal cells. Interestingly, LPS consistently appeared in the nucleus, but only occasionally appeared in the cytoplasm of the spleen, lung, and large intestine, while LTA was mainly present in the cytoplasm (Figure 3, Appendix A). We also used RNA fluorescence in situ hybridization (FISH) with EUB338 to detect bacterial RNA [32]. The results showed that bacterial 16S rRNA was present in the parenchymal cells (Figure 4).

To exclude false-positive results derived from possible contamination during sampling, sequencing, or in situ detection procedures, we conducted in vitro examinations using cultured cells. HepG2, Huh-7, HSC-T6, and Hepa1-6 cells were cultured in a germ-free environment, and LPS and LTA in cytoplasmic and nuclear extracts were detected with Western blotting. The results also showed the existence of LPS in the nuclear extracts. However, we failed to detect LTA (Figure 5, Appendix A). Together with the results of 16S rRNA gene sequencing, it can be deduced that the Gram-negative bacterial phyla Proteobacteria, Bacteroidetes, Acidobacteria, Cyanobacteria, Gemmatimonadetes, Epsilonbacteraeota, Chloroflexi, and Verrucomicrobia were the dominant nuclear bacteria and that the Gram-positive bacterial phyla Firmicutes and Actinobacteria were the dominant cytoplasmic bacteria.

### 3.2. Visceral Bacteria in Newborn Rats

Whether the visceral bacteria originate from the gut microbiome or not is a mystery. It is widely thought that the bacteria may be derived from the intestinal microbiome due to gut leakage. We thus detected bacteria within 10 h in newborn rats who had only received their first ingestion of milk. Six newborn rats were used from the same litter to reduce possible variation. For the sample quantity required for 16S rRNA gene sequencing (>1.0 g was needed), the heart, spleen, kidney, and pancreas were combined (referred to as HSKP) and analyzed together, and the intact intestinal tissue was collected with its contents instead of separating it further into the jejunum, ileum, and colon. The results showed a high abundance of bacteria in visceral organs, and the number of OTUs in visceral organs (brain, HSKP, lung, liver, and skeletal muscle) was 1634 ± 47.70 (Figure 6A). The visceral bacteria were shared with those of the milk clot and skin tissue (Figure 6B–D). However, besides *Lactobacillus*, the top 10 bacteria were completely different from those of adult male rats, and uncultured bacterium f *Muribaculaceae* and uncultured bacterium f *Lachnospiraceae* were high. The other dominant bacteria of the top 10 were *Ruminococcaceae UCG-014*, *Bacteroides*, *Weissella*, *Escherichia-Shigella*, *Collinsella*, *Alloprevotella*, and *Cellvibrio.* Compared with the milk clot and skin tissues, the visceral and intestinal bacteria had identical top 10 dominant bacteria, and smooth muscle was high in *Weissella* bacteria. Among the litter of rats, the types of bacteria were constant across individuals and visceral organs, except for the skeletal muscle and milk clot (Appendix A). Furthermore, each organ possessed exclusive bacteria with a low abundance in each individual (Appendix A). Among the top 10 dominant bacteria in visceral organs and milk clots, the abundance of Gram-negative bacteria was significantly higher than that of Gram-positive bacteria (52.40 ± 4.82% vs. 44.46 ± 5.24% and 56.68 ± 2.94% vs. 40.04 ± 2.18%, *p* < 0.05). Furthermore, the abundance of Gram-positive bacteria in milk clots was the same as that of visceral organs (*p* > 0.05), and the abundance of Gram-negative bacteria was significantly lower in visceral organs than in milk clots (*p* < 0.05).

In newborn rats, the species richness of the skin tissue and milk clot in the stomach was the same but was higher than those of the intestinal tissues (including the intestinal contents), HSKP, lung, liver, and skeletal muscle, *p* < 0.05. The species richness of liver and lung tissue was higher than that of HSKP, skeletal muscle, and intestine, *p* < 0.05. The species evenness of skin tissue was higher than that of the HSKP, brain, skeletal muscle, and lung (Appendix A). The bacteria of the milk clot, skin tissue, intestinal tissue, and visceral organs showed a clear similarity (Appendix A).

The visceral bacteria species were the same as the bacteria in the milk clot and skin tissue. It can be deduced that the bacteria are closely related to the milk from lactation and are perhaps milk-derived gut microbiota.

### 3.3. Visceral Bacteria in Pregnant Mothers, Fetuses, and Placentas

To further determine whether the visceral bacteria were exogenous from the milk or endogenous from inheritance, we further detected the bacteria of the rat fetuses and placentas. The fetus is traditionally considered sterile in mammals. We found the fetuses and placentas were also high in bacteria. The fetuses had a high abundance of bacteria with 1771 OTUs, which was the same as that of pregnant mothers. The placentas had relatively low OTUs, i.e., 1397 (Figure 7A). The fetuses and placentas shared OTUs and bacteria with the pregnant rats (Figure 7B). The top 10 bacteria were the same as the visceral bacteria of pregnant rats (Figure 7C,D). Among the top 10 dominant bacteria in the visceral organs of mother rats, fetuses, and placentas, the abundance of Gram-negative bacteria was significantly higher than that of Gram-positive bacteria (56.21 ± 7.82% vs. 40.00 ± 8.66% and 54.27 ± 2.85% vs. 41.31 ± 4.48%, *p* < 0.01). Furthermore, the abundance of Gram-positive bacteria was significantly higher than that of Gram-negative bacteria in intestinal contents of mother rats (61.17 ± 9.70% vs. 37.47 ± 9.10%, *p* < 0.01). Additionally, the abundance of Gram-positive bacteria was significantly higher in intestinal contents than in visceral organs (*p* < 0.001), and the abundance of Gram-negative bacteria was significantly higher in visceral organs than in intestinal contents (*p* < 0.001).

In each kind of organ, the abundance of dominant bacteria varied across individuals (Appendix A). In each individual, the abundance of dominant bacteria varied across organs and was generally different from the bowel microbiome (Appendix A). Although a large number of species were shared, each organ had exclusive bacteria with a low abundance (Appendix A).

The fetuses shared several dominant bacteria with newborn rats, including uncultured bacterium f *Muribaculaceae*, *Lactobacillus Ruminococcaceae UCG-014*, *Bacteroides*, uncultured bacterium f *Lachnospiraceae*, *Weissella*, and *Escherichia-Shigella*. As opposed to the newborn rats, *Collinsella*, *Alloprevotella*, and *Cellvibrio* were low in the fetuses, and *Akkermansia*, *Helicobacter*, and uncultured bacterium c subgroup 6 were high. This result suggested that the visceral bacteria may change in development from fetus to mature rat.

The placentas shared the same top 10 dominant bacteria, but there was a relatively large number of exclusive bacterial species across individuals. Fetuses from different pregnant rats showed relatively identical dominant bacterial compositions across organs and individuals (Appendix A).

According to the alpha diversity analysis, the fetuses and placentas showed nearly the same species richness and evenness as the visceral organs of pregnant rats (Appendix A), but the distribution in the fetuses and placentas was different (Appendix A).

## 4. Discussion

Cells have traditionally being regarded as the fundamental morphological unit of plants, animals, and humans [31]. The detection of bacteria within tumor cells (intratumor microbiota) [32,33,41,42,43,44,45,46,47,48] reminded us of the possibility of potential overlaps between “cells” and “animalcules”. By means of metagenomic sequencing and in situ detection, recently, we found the existence of bacteria in hepatocytes of healthy SD rats [39]. This result suggested that the bacteria in cells may be a normal life phenomenon. In this study, we further detected the existence of bacteria in parenchymal cells of various visceral organs. We performed 16S rRNA gene sequencing detection in rats and repeated the in situ detection more than three times. Although it is surprising, the results supported the fact that bacteria were located in parenchymal cells in rats.

Several studies have reported the positive detection of 16S rRNA genes in tissues and circulation [36,49,50,51,52], but in situ detection was less frequently performed. Moreover, these studies considered bacteria from the gut microbiome, and a comparison of the tissue bacteria with the gut microbiome was lacking. In this study, in situ detection confirmed the tissue bacteria were in fact intracellular. Interestingly, Gram-negative bacteria were located in the nucleus, while Gram-positive bacteria were located in the cytoplasm. By comparing the visceral microbiome with the gut microbiome, we showed that the cellular microbiome in visceral organs differed from that of the gut microbiome structurally and functionally, despite containing the same species. The number of OTUs, richness, evenness, and similarity in visceral bacteria were even higher than those of the small intestine contents and feces. A high abundance of 16S rRNA genes were also detected in newborn rats and fetuses, confirming that the visceral bacteria had no relationship with the gut microbiome. This challenges the concept that the tissue bacteria came from the gut microbiome.

In male adult rats, the visceral bacteria showed higher similarity and were more consistent among rats than the gut microbiome. Only very few annotated bacteria were exclusive to the brain, heart, lung, kidney, and spleen, suggesting that visceral bacteria possessed universal characteristics in the rats. However, the dominant bacteria differed across organs within an individual, and differed across individuals for a given organ, suggesting the uniqueness of visceral bacteria. These features were also shown in pregnant rats (Appendix A). Owing to sampling difficulties, organs in newborn rats and fetuses could not be detected individually. However, they all showed a high abundance of 16S rRNA genes. Results showed that the visceral bacteria in newborn rats were similar to those of the milk clot. Milk bacteria have been reported recently [53,54,55,56,57,58,59], and thus can be a source of an individual’s gut microbiome [60,61,62]. The visceral bacteria in the fetuses had the same abundance and alpha diversity as their mothers; however, the beta diversity analysis results differed from their mothers (Appendix A), suggesting the fetal and placental bacteria were distinct and perhaps from the initial zygote cells. Placental and fetal bacteria have also been reported recently [63,64,65,66,67]. According to these reports and our result of the beta diversity analysis, it can be deduced that the placental microbiome should be considered a part of the embryo rather than a part of the mother’s gut microbiome.

The intratumor microbiome, also known as the thanatomicrobiome in decayed mammalian visceral organs, showed time-, organ-, and sex-dependent features. Changes in the signature microorganisms of each organ showed potential as markers in the field of forensic medicine for deducing the time of death [68,69,70]. It was hypothesized that the thanatomicrobiome originated from the gut and lungs, which were exposed to the external environment. However, the mechanisms by which these bacteria gain access to visceral organs and spread from the gut and lungs to other visceral organs after death remain a mystery. The top 10 dominant compositions showed overlap with intracellular microbial communities, including phyla such as Firmicutes, Actinobacteria, Bacteroidetes, Proteobacteria, and Acidobacteria, among others [68,69]. This suggests that the cellular microbiome could be a plausible source of the thanatomicrobiome, originating from parenchymal cells that are normally protected by living cells and contributing to host decay after death (autolysis). Variations in the thanatomicrobiome communities may also provide valuable insights into the individual’s health status during their lifetime, such as tumor and metabolic diseases.

One limitation of this study is the lack of more in-depth microbiological, morphological, and functional validation that could provide comprehensive evidence of the presence of tissue and cellular microbiota. In addition, the experimental results do not provide a detailed explanation of the source of intracellular microbiota in parenchymal cells.

In conclusion, this article reported that cellular microbiome resided in parenchymal cells as inherent inhabitants in visceral organs of rats. They were cellular intrinsic inhabitants rather than translocated through gut leakage or the fetal blood barrier. Further study is needed to identify these endogenous intracellular bacteria in different organs.

## Figures and Tables

**Figure 1 microorganisms-12-01333-f001:**
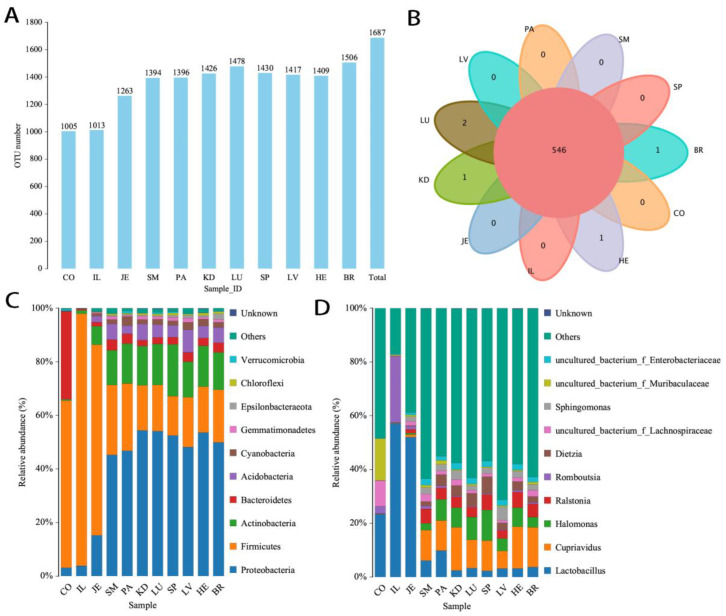
Bacteria in visceral organs (*n* = 6). (**A**) Number of OTUs of samples; (**B**) Venn diagram of shared OTUs in visceral organs and the gut microbiome; (**C**) species distribution (phyla); and (**D**) species distribution (genera).

**Figure 2 microorganisms-12-01333-f002:**
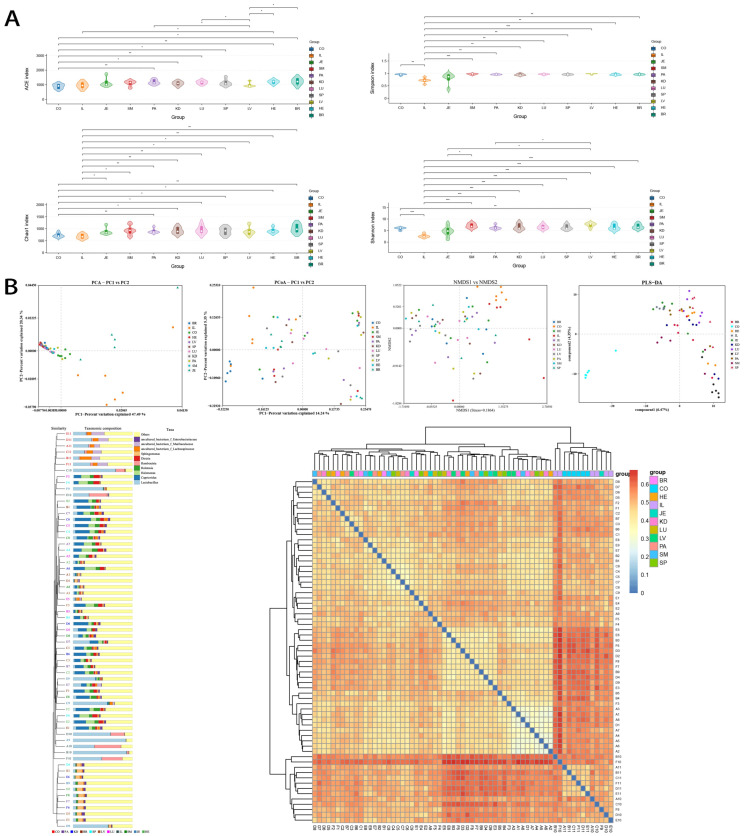
Diversity features of visceral bacteria (*n* = 6). (**A**) Alpha diversity analysis. Visceral bacteria showed higher richness and abundance. * *p* < 0.05; ** *p* < 0.01; *** *p* < 0.001. Simpson indices were calculated as 1−∑*p_i_*^2^. (**B**) Beta diversity analysis. A to F represent six rats. In addition to the color labeling, organs are also numbered: 1, brain (BR); 2, heart (HE); 3, liver (LV); 4, spleen (SP); 5, lung (LU); 6, kidney (KD); 7, pancreas (PA); 8, skeletal muscle (SM); 9, jejunum (JE); 10, ileum (IL); and 11, colon (CO).

**Figure 3 microorganisms-12-01333-f003:**
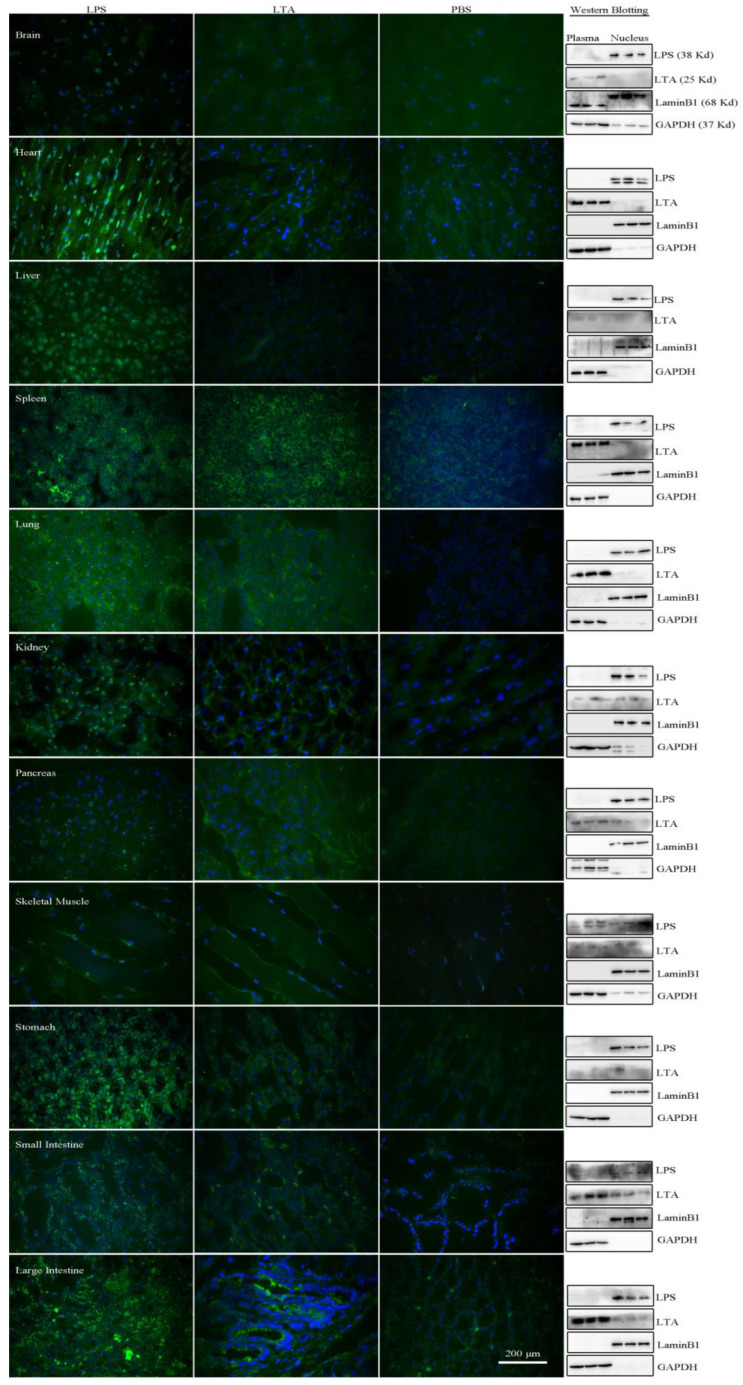
LPS and LTA detected in parenchymal cells. Visceral organs were positive for LPS and LTA, with LPS located in the nuclei showing granular fluorescence (positive control with *E. coli*.), while the LTA was located mainly in the cytoplasm with uniform fluorescence (positive control with *Staphylococcus aureus*). Green, FITC labeling; blue, DAPI counterstaining. LPS: lipopolysaccharide. LTA: lipoteichoic acid. FITC: fluorescein isothiocyanate. Bar = 200 μm. FITC labeling and DAPI counterstaining images were merged using the software NIS-Elements D (5.20.00, Build 1423). The exposure time of the immunofluorescence observation was 700 ms. The quantity of protein samples was 30 μg, except for the heart tissue, where it was 40 μg.

**Figure 4 microorganisms-12-01333-f004:**
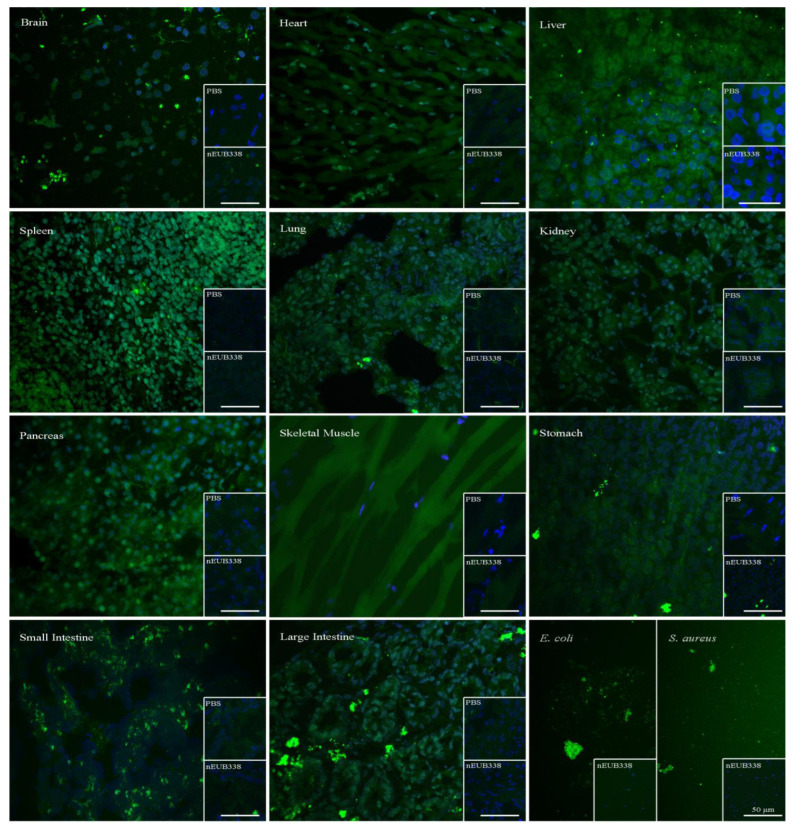
Positive reaction of EUB338 probes in parenchymal cells. Parenchymal cells were EUB338 probe-positive in the nuclei and cytoplasm. *E. coli* and *S. aureus* were used as the positive controls, and PBS and nEUB338 were used as blank and the negative control, respectively. Green, FITC labeling; blue, DAPI counterstaining. PBS: phosphate-buffered saline. FITC: fluorescein isothiocyanate. Bar = 50 μm. FITC labeling and DAPI counterstaining images were merged with software NIS-Elements D. The exposure time of the immunofluorescence observations was 350 ms.

**Figure 5 microorganisms-12-01333-f005:**
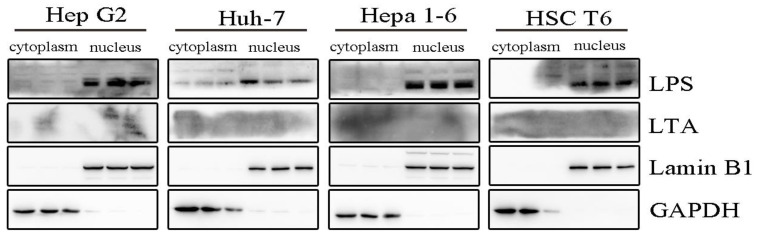
LPS was positively detected in cultured cell lines. LPS showed a positive reaction in Hep G2, Huh-7, Hepa1-6, and HSC T6 cells in Western blotting and was mainly located in the nucleus. LTA failed to be detected. LPS: lipopolysaccharide; LTA: lipoteichoic acid. The quantity of protein samples was 30 μg.

**Figure 6 microorganisms-12-01333-f006:**
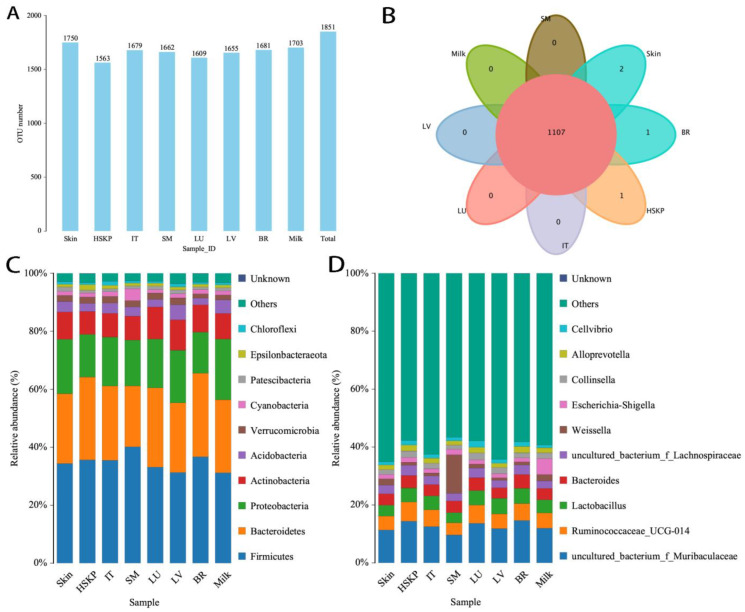
Bacteria in newborn rats (within 10 hours of birth). (**A**) Number of OTUs of samples; (**B**) Venn diagram of shared OTUs in visceral organs of newborn rats; (**C**) top 10 species distribution (phylum); and (**D**) top 10 species distribution (genus).

**Figure 7 microorganisms-12-01333-f007:**
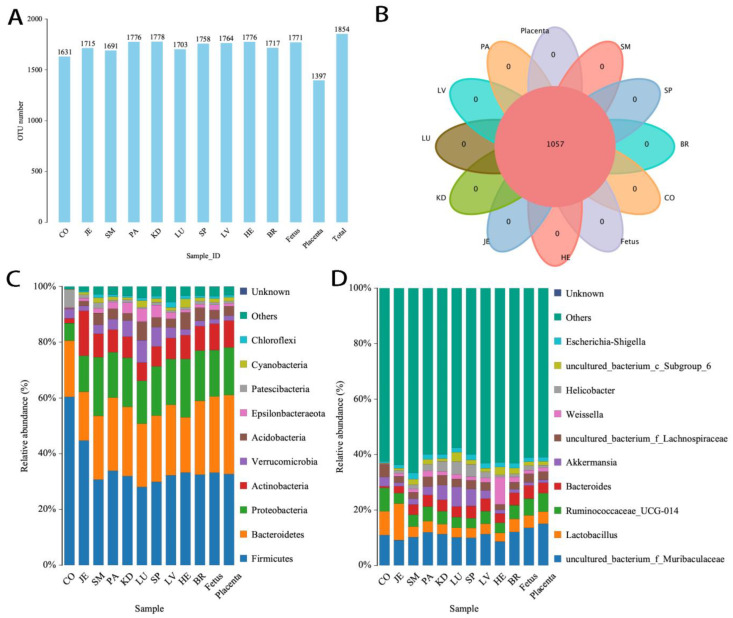
Bacteria in visceral organs of pregnant rats and their fetuses (*n* = 5). (**A**) Number of OTUs of samples; (**B**) Venn diagram of shared OTUs in visceral organs of pregnant rats and their fetuses and placentas; (**C**) species distribution (phylum); and (**D**) species distribution (genus).

## Data Availability

The raw datasets generated during the current study are available in the NCBI repository (https://www.ncbi.nlm.nih.gov/) (accessed on 24 April 2022 and 7 February 2023) under the following BioProject accession numbers: PRJNA831335, PRJNA857281, and PRJNA857328. Raw data from Figure 1, Figure 2, Figure 3, Figure 4, Figure 5, Figure 6, Figure 7 and Appendix A have been deposited in Mendeley Data at https://data.mendeley.com/preview/htdtvy5mm7?a=07eeeb8b-0323-4bfd-9b28-2902f0796444 (accessed on 5 September 2022).

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
