# Peer review of "The Cellular Microbiome of Visceral Organs: An Inherent Inhabitant of Parenchymal Cells"

_microorganisms, 2024, doi:10.3390/microorganisms12071333_

Round 1

Reviewer 1 Report

Comments and Suggestions for Authors

The paper presented for review refers to the existence of tissue bacteria in tumor and normal tissues and their research, which proved the cellular microbiome in parenchymal cells of visceral organs as inherent inhabitants.

In the abstract, the Authors pointed out the structure of the cell and that 164 rRNA gene sequencings led to knowledge of existence of tissue bacteria. They also reported about their previous research on cellular microbiome which resides in hepatocytes and their further research presented in this paper about cellular microbiome in the parenchymal cells of visceral organs as inherent inhabitants. They then submit information about the test method, the research tool, and present the results of the study. The range of keywords covers all the issues discussed in the article.

In the introduction, the authors point out that microorganisms lead to infection and diseases and presented the early discovery by van Leeuwenhoek. They also presented that beside infectious diseases there are many non-infectious diseases that become major threats to health and that molecular and cellular research is conducted to understand mechanism and prevention of them.  Development of 16S rRNA gene sequencing is a breakthrough in bacteria identification.

The aim of the Authors' study was to show that bacteria are inherent inhabitants of normal parenchymal cells.

Material and methods: Authors described materials and methods they used starting from animals to cell and culturing, sample collection, 16S rRNA gene sequencing, immunofluorescence assays, Western Blotting.

Specific points:

I suggest that this section comes before Results.
Line 440-444: Can you provide the number of animals you used in the study?

Results are very well presented, section about visceral bacteria in male adult sd rats followed by section about visceral bacteria in newborn and section about visceral bacteria in pregnant mothers, fetus, and placenta. The results of the Author's own research are presented in the article on 7 main figures and 19 supplementary figures.

The discussion is extensive, and the Authors draw attention to many important issues of their own study in correlation with the results obtained by other authors. The Authors also referred that several studies have reported the positive detection of 16S rRNA genes in tissues and circulation but in situ detection was not so represented in research. They showed in this study, that tissue bacteria is intracellular confirmed by in situ detection. Furthermore they discussed their results  and compared them to other researches.

Specific point:

378-385 This section should be in Results.  

343-345: “We performed 16S rRNA gene sequencing 343 detection in rats (www.biomarker.com.cn, BMK200916–AC763–0101 and BMK210512–344 AJ541–ZX01-0101)”

This part: “ (www.biomarker.com.cn, BMK200916–AC763–0101 and BMK210512–344 AJ541–ZX01-0101)” is not for Discussion, it is part of Methodology

Same comment is for Line 377

In their conclusions, the Authors emphasized the importance of conducting further studies to identify intracellular bacteria in different organs as they reported that cellular microbiome resides in parenchymal cells as intrinsic inhabitants.

The results obtained by the authors have a practical dimension and can be the basis for further research.

Note: Can you  point out on any limitation of your study?

The analysis of the topics raised by the Authors has been presented in a clear and coherent manner. All items of scientific literature (82) are up-to-date and can be found in the text of the article. The language of the work is understandable and easy to read.

I rate this study very well in terms of content and recommend it for publication after minor additions indicated in the review.

Author Response

Response to Reviewer 1 Comments

1. Summary

Thank you very much for taking the time to review this manuscript. Please find the detailed responses below and the corresponding revisions and corrections highlighted or marked with track changes in the re-submitted files.

2. Questions for General Evaluation

Reviewer’s Evaluation

Response and Revisions

Does the introduction provide sufficient background and include all relevant references?

Yes

Thank you for the affirmation.

Are all the cited references relevant to the research?

There is no comment.

Is the research design appropriate?

Yes

Thank you for the affirmation.

Are the methods adequately described?

Yes

Thank you for the affirmation.

Are the results clearly presented?

Yes

Thank you for the affirmation.

Are the conclusions supported by the results?

Yes

Thank you for the affirmation.

3. Point-by-point Responses to Comments and Suggestions for Authors

Comment 1:

The paper presented for review refers to the existence of tissue bacteria in tumor and normal tissues and their research, which proved the cellular microbiome in parenchymal cells of visceral organs as inherent inhabitants.

In the abstract, the Authors pointed out the structure of the cell and that 164 rRNA gene sequencings led to knowledge of existence of tissue bacteria. They also reported about their previous research on cellular microbiome which resides in hepatocytes and their further research presented in this paper about cellular microbiome in the parenchymal cells of visceral organs as inherent inhabitants. They then submit information about the test method, the research tool, and present the results of the study. The range of keywords covers all the issues discussed in the article.

In the introduction, the authors point out that microorganisms lead to infection and diseases and presented the early discovery by van Leeuwenhoek. They also presented that beside infectious diseases there are many non-infectious diseases that become major threats to health and that molecular and cellular research is conducted to understand mechanism and prevention of them. Development of 16S rRNA gene sequencing is a breakthrough in bacteria identification.

The aim of the Authors' study was to show that bacteria are inherent inhabitants of normal parenchymal cells.

Response 1: Thank you for your affirmation and careful reading of our manuscript.

Comment 2:

Material and methods: Authors described materials and methods they used starting from animals to cell and culturing, sample collection, 16S rRNA gene sequencing, immunofluorescence assays, Western Blotting.

Specific points:

I suggest that this section comes before Results.

Response 2: We agree and have, accordingly, moved this section before the Results. This change can be found on pages 3–5 in the revised manuscript.

Comments 3: Line 440-444: Can you provide the number of animals you used in the study?

Response 3: We have added the number of animals used in this study to the manuscript. The relevant information can be found on page 3, in “2.1 Animals.” We have marked this change in red.

Comment 4: Results are very well presented, section about visceral bacteria in male adult sd rats followed by section about visceral bacteria in newborn and section about visceral bacteria in pregnant mothers, fetus, and placenta. The results of the Author's own research are presented in the article on 7 main figures and 19 supplementary figures.

Response 4: Thank you for your affirmation.

Comment 5: The discussion is extensive, and the Authors draw attention to many important issues of their own study in correlation with the results obtained by other authors. The Authors also referred that several studies have reported the positive detection of 16S rRNA genes in tissues and circulation but in situ detection was not so represented in research. They showed in this study, that tissue bacteria is intracellular confirmed by in situ detection. Furthermore they discussed their results and compared them to other researches.

Specific point:

378-385 This section should be in Results.

Response 5.1: We agree and have, accordingly, removed this statement, which is redundant with the results. This change can be found on page 9, paragraph 3, lines 13–14, and has been marked in red in the revised manuscript.

343-345: “We performed 16S rRNA gene sequencing 343 detection in rats (www.biomarker.com.cn, BMK200916–AC763–0101 and BMK210512–344 AJ541–ZX01-0101)”

This part: “ (www.biomarker.com.cn, BMK200916–AC763–0101 and BMK210512–344 AJ541–ZX01-0101)” is not for Discussion, it is part of Methodology.

Response 5.2: We agree and have, accordingly, moved “(www.biomarker.com.cn, BMK200916–AC763–0101 and BMK210512–344 AJ541–ZX01-0101)” to Materials and Methods Section 2.4, paragraph 2. This change can be found in paragraph 2 on page 4 in the revised manuscript; we have marked this change in red.

Same comment is for Line 377

Response 5.3: We agree and have, accordingly, removed “(#HM6011, #HM2048; Hycult)” on Line 377 due to this information having been mentioned in Materials and Methods Section 2.7.

Comment 6: In their conclusions, the Authors emphasized the importance of conducting further studies to identify intracellular bacteria in different organs as they reported that cellular microbiome resides in parenchymal cells as intrinsic inhabitants.

The results obtained by the authors have a practical dimension and can be the basis for further research.

Response 6: Thank you for your affirmation.

Comment 7: Note: Can you point out on any limitation of your study?

Response 7: We have added a discussion of limitations to this study. The relevant information can be found on page 10, paragraph 3, lines 7–11, and we have marked this change in red.

Comment 8: The analysis of the topics raised by the Authors has been presented in a clear and coherent manner. All items of scientific literature (82) are up-to-date and can be found in the text of the article. The language of the work is understandable and easy to read.

I rate this study very well in terms of content and recommend it for publication after minor additions indicated in the review.

Response 8: Thank you for your affirmation. We have revised the manuscript according to the comments in the review and have added further information as needed.

4. Response to Comments on the Quality of English Language

Point 1: English language fine. No issues detected.

Response 1: Thank you for your affirmation.

5. Additional clarifications

We have renumbered the sections and references due to the Materials and Methods section being moved before the Results, and we have also inserted several new references.

Reviewer 2 Report

Comments and Suggestions for Authors

Review of the paper entitled „The Cellular Microbiome of Visceral Organs: An Inherent Inhabitant of Parenchymal Cells” by Xiao Wei Sun, Hua Zhang, Xiao Zhang, Wen Min Gao, Cai Yun Zhou, Xuan Xuan Kou, Jing Xin Deng and Jian Gang Zhang

    The Authors put forward a very original and bold hypothesis that bacteria are inherent inhabitants of normal parenchymal cells.

     The Authors should improve the Materials and Methods section. How many animals were there in total, how many animals were in each group, how old were the animals, etc. How were the animals killed, how exactly was the tissue collection carried out?

     The thanatomicrobiome refers to the microbial communities that colonize the internal organs of a corpse during the post-mortem period. It has been shown that in the thanatomicrobiome of liver the phylum Firmicutes and Proteobactteria dominate.

Are the Authors sure that bacteria are inherent inhabitants of normal, living parenchymal cells? That it's not thanatomicrobiome? Please leave a short comment in the Discussion on this topic.

Author Response

Response to Reviewer 2 Comments

1. Summary

Thank you very much for taking the time to review this manuscript. Please find the detailed responses below and the corresponding revisions and corrections highlighted or marked with track changes in the re-submitted files.

2. Questions for General Evaluation

Reviewer’s Evaluation

Response and Revisions

Does the introduction provide sufficient background and include all relevant references?

Yes

Thank you for the affirmation.

Are all the cited references relevant to the research?

There is no comment.

Is the research design appropriate?

Yes

Thank you for the affirmation.

Are the methods adequately described?

Can be improved

We have added some key information according to the comments.

Are the results clearly presented?

Yes

Thank you for the affirmation.

Are the conclusions supported by the results?

Can be improved

We have added some key information according to the comments.

3. Point-by-point Responses to Comments and Suggestions for Authors

Comment 1:

Review of the paper entitled ”The Cellular Microbiome of Visceral Organs: An Inherent Inhabitant of Parenchymal Cells” by Xiao Wei Sun, Hua Zhang, Xiao Zhang, Wen Min Gao, Cai Yun Zhou, Xuan Xuan Kou, Jing Xin Deng and Jian Gang Zhang

The Authors put forward a very original and bold hypothesis that bacteria are inherent inhabitants of normal parenchymal cells.

The Authors should improve the Materials and Methods section. How many animals were there in total, how many animals were in each group, how old were the animals, etc. How were the animals killed, how exactly was the tissue collection carried out?

Response 1: Thank you for pointing this out. We agree with this comment. Therefore, we have added information on the animals used to the Materials and Methods section. The relevant information regarding the animal number and body weight can be found on page 3, “2.1 Animals”; the method for animal sacrifice can be found on page 3, “2.3. Sample Collection and Contamination Avoidance;” and the detailed information on tissue collection can be found on page 3, paragraph 9. We have marked these changes in red.

Comment 2: The thanatomicrobiome refers to the microbial communities that colonize the internal organs of a corpse during the post-mortem period. It has been shown that in the thanatomicrobiome of liver the phylum Firmicutes and Proteobactteria dominate.

Are the Authors sure that bacteria are inherent inhabitants of normal, living parenchymal cells? That it's not thanatomicrobiome? Please leave a short comment in the Discussion on this topic.

Response 2: Thank you for pointing this out. In this study, we performed 16S rRNA gene sequencing detection in rats with five to six repeated samples and two repeated detections, and the results supported the fact that bacteria were located in parenchymal cells in rats. We performed in situ FISH and IF on visceral organ tissues more than three times and found that, compared with the negative controls, the parenchymal cells of visceral organs showed positive reactions to LPS, LTA, and the probe EUB 338. To verify the cellular location of bacteria in cells, we further performed western blotting on the cytoplasm and nuclear extracts of tissues and cultured HepG2, Huh-7, Hepa1-6, and HSC-T6 (rat hepatic stellate) cells. The results showed that LPS was located in the nucleus (Figure 5). The LTA was not detected in cultured cells. These results indicate that the bacteria detected in visceral organs were intracellular inhabitants in normal living parenchymal cells.

All samples were collected immediately from killed animals rather than from corpses, so the intracellular microbiome could not be considered the thanatomicrobiome. However, due to the overlapping annotation of the intracellular microbiome with the thanatomicrobiome, there may be a relationship between them. We have made a short comment regarding this point in the Discussion on page 10, paragraph 2, and have added three related references.

4. Response to Comments on the Quality of English Language

Point 1: I am not qualified to assess the quality of English in this paper

Response 1: Thank you for your careful review.

5. Additional clarifications

None.

Reviewer 3 Report

Comments and Suggestions for Authors

The manuscript presents a very intriguing, although insufficiently documented, topic: the presence of intracellular and intranuclear bacteria as part of the normal mammalian microbiome. Although the research methods are sensitive and up-to-date, the sample size (i.e., the number of tissue types) is small, most of the samples are cell lines and the controls are not adequate. As such, the scientific conclusions of this paper are highly speculative, unrealistic and not sustained by the research results. Although the research has some merit, highlighting a new field of microbiome research, the data provided by this research is not convincing.

Comments on the Quality of English Language

The English language of the manuscript is adequate, however some retouching is necessary. Also, the 'Introduction' and the "Discussion' sections are redundant, should be condensed and the stories about the history of microscopy should be removed entirely as they do not pertain directly to the understanding of this research. However, more info and references should be added regarding the established facts on the intracellular microbiome, like the intratumoral microbiome. Abbreviations, like LPS and LTA, should be defined from the beginning, in the Abstract and the Introduction, as the unaware reader has no idea what those could be.

Author Response

Response to Reviewer 3 Comments

1. Summary

Thank you very much for taking the time to review this manuscript. Please find the detailed responses below and the corresponding revisions and corrections highlighted or marked with track changes in the re-submitted files.

2. Questions for General Evaluation

Reviewer’s Evaluation

Response and Revisions

Does the introduction provide sufficient background and include all relevant references?

Must be improved

We have added some key information according to the comments.

Are all the cited references relevant to the research?

There is no comment.

Is the research design appropriate?

Must be improved

We have added some key information according to the comments.

Are the methods adequately described?

Can be improved

We have added some key information according to the comments.

Are the results clearly presented?

Must be improved

We have added some key information according to the comments.

Are the conclusions supported by the results?

Must be improved

We have added some key information according to the comments.

3. Point-by-point response to Comments and Suggestions for Authors

Comment 1: The manuscript presents a very intriguing, although insufficiently documented, topic: the presence of intracellular and intranuclear bacteria as part of the normal mammalian microbiome. Although the research methods are sensitive and up-to-date, the sample size (i.e., the number of tissue types) is small, most of the samples are cell lines and the controls are not adequate. As such, the scientific conclusions of this paper are highly speculative, unrealistic and not sustained by the research results. Although the research has some merit, highlighting a new field of microbiome research, the data provided by this research is not convincing.

Response 1: Thank you for pointing this out. In this study, we performed 16S rRNA gene sequencing detection in rats with five to six repeated samples and two repeated detections (www.biomarker.com.cn, BMK200916–AC763–0101 and BMK210512–AJ541–ZX01-0101). The results indicated that bacteria were located in parenchymal cells in rats.

We performed in situ FISH and IF on visceral organ tissues more than three times, and we found that, compared with the negative controls, the parenchymal cells of visceral organs showed positive reactions to LPS, LTA, and the probe EUB 338.

To verify the cellular location of bacteria in cells, we further performed western blotting on the cytoplasm and nuclear extracts of tissues and cultured HepG2, Huh-7, Hepa1-6, and HSC-T6 (rat hepatic stellate) cells. The results showed that LPS was located in the nucleus. These results indicate that the bacteria detected in visceral organs were intracellular inhabitants in normal living parenchymal cells.

Because the investigated animals and cell lines were not given any treatment, there was no treatment control group set in this study, except for the technical control. Due to the uniformity of diet and biological characteristics of SD rats, 5–6 animals in each group are usually selected as biological duplicates in animal experiments. In this study, we selected a similar number of samples. Meanwhile, according to the alpha diversity and beta diversity analysis results, the visceral microbiome exhibited obvious uniformity of species; 5–6 samples could reflect these features of diversity in the visceral microbiome.

Comment 2:

Also, the 'Introduction' and the "Discussion' sections are redundant, should be condensed and the stories about the history of microscopy should be removed entirely as they do not pertain directly to the understanding of this research.

Response 2: Thank you for pointing this out. We agree with this comment. We have, accordingly, deleted redundant content in the Introduction and Discussion, and the review of the history of microscopy has been completely removed.

Comment 3:

More info and references should be added regarding the established facts on the intracellular microbiome, like the intratumoral microbiome.

Response 3: Thank you for pointing this out. We agree with this comment. We have, accordingly, added 10 leading references (50, 51, and 56–63) about the intratumoral microbiome.

Comment 4:

Abbreviations, like LPS and LTA, should be defined from the beginning, in the Abstract and the Introduction, as the unaware reader has no idea what those could be.

Response 4: Thank you for pointing this out. We agree with this comment. We have, accordingly, defined all abbreviations, including LPS, LTA, and 16S rRNA, in the Abstract and the Introduction, and we have marked the changes in red. The modifications can be found on page 1, lines 2 and 14 (for the Abstract), as well as page 2, paragraph 2, line 4 (for the Introduction).

4. Response to Comments on the Quality of English Language

Point 1: Moderate editing of English language required

The English language of the manuscript is adequate, however some retouching is necessary.

Response 1: Thank you for pointing this out. We have asked LetPub (www.letpub.com) to make further modifications.

5. Additional clarifications

None.

Round 2

Reviewer 2 Report

Comments and Suggestions for Authors

The authors have significantly improved their paper. The authors have taken into account my comments and suggestions.

Author Response

Comment 1:

The authors have significantly improved their paper. The authors have taken into account my comments and suggestions.

Response 1: Thank you for your affirmation.

4. Comment 2. Response to Comments on the Quality of English Language

Point 1: I am not qualified to assess the quality of English in this paper

Response 1: Thank you for your careful review.

Reviewer 3 Report

Comments and Suggestions for Authors

The manuscript has been improved somehow but it still requires major rewriting of the Introduction and Discussion sections. They should be compressed and general irrelevant information (such has the historical microscopy aspects) should be removed.

Comments on the Quality of English Language

English language requires some average tweaking in order to improve the clarity of the presentation.

Author Response

Comment 1: The manuscript has been improved somehow but it still requires major rewriting of the Introduction and Discussion sections. They should be compressed and general irrelevant information (such has the historical microscopy aspects) should be removed.

Response 1: Thank you for pointing this out. We agree with this comment. We have, accordingly, deleted 307 words and 10 references(references 11-15, 17, 20, 26, 46, 47) in Introduction, and 640 words and 10 references (67-76) in Discussion., and the the history of microscopy and references (1-3) has been completely removed. The corresponding revisions and corrections highlighted or marked with track changes in the re-submitted files.

4. Response to Comments on the Quality of English Language

Point 1: Moderate editing of English language required

English language requires some average tweaking in order to improve the clarity of the presentation.

Response 1: Thank you for pointing this out. We have asked LetPub (www.letpub.com) to make further modifications. The corresponding revisions and corrections highlighted or marked with track changes in the re-submitted files. The certification can be found in uploaded file.

5. Additional clarifications

We have renumbered the references due to the major revision of Introduction and Discussion.
